# Discovery of Long Non-Coding RNA MALAT1 Amplification in Precancerous Colorectal Lesions

**DOI:** 10.3390/ijms23147656

**Published:** 2022-07-11

**Authors:** Anna Siskova, Jan Kral, Jana Drabova, Klara Cervena, Kristyna Tomasova, Jiri Jungwirth, Tomas Hucl, Pavel Kohout, Sandra Summerova, Ludmila Vodickova, Pavel Vodicka, Veronika Vymetalkova

**Affiliations:** 1Department of Molecular Biology of Cancer, Institute of Experimental Medicine of the Czech Academy of Sciences, Videnska 1083, 142 00 Prague, Czech Republic; klara.cervena@iem.cas.cz (K.C.); kristyna.tomasova@iem.cas.cz (K.T.); ludmila.vodickova@iem.cas.cz (L.V.); pavel.vodicka@iem.cas.cz (P.V.); 2Institute of Biology and Medical Genetics, 1st Faculty of Medicine, Charles University and General University Hospital in Prague, Albertov 4, 128 00 Prague, Czech Republic; 3Clinic of Hepatogastroenterology, Institute for Clinical and Experimental Medicine, Videnska 1958/9, 140 21 Prague, Czech Republic; jan.kral@ikem.cz (J.K.); tomas.hucl@ikem.cz (T.H.); 4Department of Biology and Medical Genetics, 2nd Faculty of Medicine, Charles University and University Hospital Motol, V Uvalu 84, 150 06 Prague, Czech Republic; jana.drabova@fnmotol.cz; 5Biomedical Centre, Faculty of Medicine in Pilsen, Charles University, Alej Svobody 76, 323 00 Pilsen, Czech Republic; 6Institute of Physiology, 1st Faculty of Medicine Charles University, Katerinska 1660/32, 121 08 Praha, Czech Republic; jiri.jungwirth@kliniken-nordoberpfalz.ag; 7Department of Gastroenterology, Libera Scientia, Lucni 7a, 130 00 Prague, Czech Republic; 8Department of General, Visceral and Thoracic Surgery, Klinikum Weiden, Söllnerstraße 16, 92637 Weiden, Germany; 9Department of Internal Medicine, 3rd Faculty of Medicine University and Faculty Thomayer Hospital Prague, Ruska 87, 100 00 Prague, Czech Republic; pavel.kohout@ftn.cz (P.K.); sandra.summerova@ftn.cz (S.S.)

**Keywords:** colorectal cancer, adenomas, array comparative genomic hybridization, long non-coding RNA, MALAT1

## Abstract

A colorectal adenoma, an aberrantly growing tissue, arises from the intestinal epithelium and is considered as precursor of colorectal cancer (CRC). In this study, we investigated structural and numerical chromosomal aberrations in adenomas, hypothesizing that chromosomal instability (CIN) occurs early in adenomas. We applied array comparative genomic hybridization (aCGH) to fresh frozen colorectal adenomas and their adjacent mucosa from 16 patients who underwent colonoscopy examination. In our study, histologically similar colorectal adenomas showed wide variability in chromosomal instability. Based on the obtained results, we further stratified patients into four distinct groups. The first group showed the gain of *MALAT1* and *TALAM1*, long non-coding RNAs (lncRNAs). The second group involved patients with numerous microdeletions. The third group consisted of patients with a disrupted karyotype. The fourth group of patients did not show any CIN in adenomas. Overall, we identified frequent losses in genes, such as *TSC2*, *COL1A1*, *NOTCH1*, *MIR4673*, and *GNAS*, and gene gain containing *MALAT1* and *TALAM1*. Since long non-coding RNA MALAT1 is associated with cancer cell metastasis and migration, its gene amplification represents an important event for adenoma development.

## 1. Introduction

Colorectal adenomas are abnormally growing intestinal epithelium that originates from colon crypts, where the stem cell profiling and differentiation have been disrupted due to changes in DNA. Some adenomas display altered genetic background (chromosomal rearrangement, mutation in tumor suppressor genes and oncogenes or epigenetic modifications) that predispose them to develop into colorectal cancer (CRC) [1]. CRC is the third most common cancer and second leading cause of death due to cancer worldwide [2]. Early detection of adenomas due to their potential to evolve into CRC substantially improves the patient’s prognosis. For instance, CRC with stage I exhibits five-year overall survival (OS) of 90%, while the diagnosis in the late stages, III and IV, predicts OS of only 13% [3]. The risk of cancer development from adenomas increases with age, from 25% at age 55 years to 40% at age 80 years [4]. The process of adenoma transformation into cancer can last 5 to 10 years, depending on the accumulation of genetic and epigenetic alterations [5]. Malignant progression of adenoma to cancer is a multifactorial process that includes chromosomal instability (CIN), microsatellite instability (MSI), the influence of epigenetic factors, such as methylation of CpG islands (CIMP), and mutations in driver genes [6]. Mutations in driver genes responsible for adenoma–carcinoma progression have been identified by human genome profiling, and these patterns are referred to as driver mutations. Among the well-known genes whose driver mutations contribute to CRC development are, for example, *APC*, *KRAS*, *BRAF*, *TP53*, *PTEN*, *SMAD4*, *GNAS*, *NOTCH1*, *POLD1*, *POLE* and *MUTYH* [7,8,9,10,11].

CIN is caused by aberrant segregation of chromosomes during mitosis and is found in 65–70% of sporadic CRC cases. CIN is defined by losses or gains of loci on short or long arms of chromosomes or even losses or gains of whole chromosomes [12]. However, CIN is not a frequent subject of research in colorectal adenomas. We assume that CIN will be present already in adenomas. Clonal expansion of aberrant cells forms the basis for the intertumoral heterogeneity within the adenoma and consequently within the tumor. Therefore, each adenoma of an individual may display a unique genetic background [13]. The genetically diverse cell population results in somatic mosaicism, a common phenomenon in tumors, while in adenomas it is an unexplored area. Somatic mosaicism in tumors or adenomas is understood as the occurrence two or more genetically distinct cell populations within one tissue [14]. We assume, based on the 70% incidence of CIN in tumors, that CIN may appear already in the precancerous lesions.

Epigenetics has been revealed to be a major player in current cancer research. Regulation of gene expression can be influenced by non-coding RNAs (long or micro RNAs). Transcripts of long non-coding RNAs have more than 200 nucleotides and may function as tumor suppressors as well as oncogenes during cancer development [15]. Metastasis-associated lung adenocarcinoma transcript 1 (MALAT1) was firstly described in relation to lung cancer aggressiveness [16]. Long non-coding RNA MALAT1 affects cell proliferation by upregulation of the Wnt/β-catenin signaling pathway, regulates transcription and post-transcriptional modification of many genes, and acts as a microRNA sponge [17]. Our hypothesis is that amplification of MALAT1 already in adenoma tissue could be a precursor of cancer development.

This study aimed to describe chromosomal aberrations in colorectal adenoma tissues with similar histological background and clinical characteristics. This study stands out by demonstrating the array-based comparative genomic hybridization (aCGH) method over a wider number of precancerous colorectal stages using DNA from fresh frozen samples, in contrast to other studies dealing mainly with colorectal tumors using DNA predominantly from formalin-fixed paraffin-embedded (FFPE) samples. This study provides a novel insight into CIN in the precancerous stages, a poorly explored area that is nonetheless crucial to understanding tumorigenesis.

## 2. Results

The aCGH method was successfully performed on all 16 pairs of samples. Enzymatic labeling of DNA was fully achieved in the same 1:1 concentration ratio within each pair of samples. Derivative log ratio standard deviation (DLRSD) ranged from 0.1 to 0.2, which reflected low probe-to-probe log-ratio noise. 

All 16 pairs of adenoma tissue samples showed varying degrees of chromosome-level instability. After a more detailed examination of these profiles, we observed that certain changes were repeated many times. Based on these obtained data, we were able to identify several groups gathering similar characteristics, and these served as a basis for creation of our four groups. Briefly, a general feature of the first group was the gain in adenoma tissue of chromosome 11, 11q13.1, encoding long non-coding RNA (lncRNA) *MALAT1* and its antisense transcript *TALAM1*, in patients P1, P2, P3, P5, and P16. The second group consisted of patients with numerous chromosomal microdeletions in adenoma tissues compared to the adjacent tissue with no aberrations in the region encoding for *MALAT1* or *TALAM1* (P6, P7, and P10). The third group included those patients with a disrupted karyotype with many losses and gains. The only common feature of this group was the relatively young age of the patients (P4, P5, and P16). In the last group, no differences were found between adenoma and adjacent tissue (P8, P9, P11, P12, P13, P14, and P15). Patients P5 and P16 overlapped between the first and third groups. The detailed results are described in the following subsections.

In addition, mosaicism was also detected in all nine samples with aberrations (P1, P2, P3, P4, P5, P6, P7, P10, P16) in adenoma tissues compared to the adjacent tissues. The percentage of mosaicism for each aberration in individual patients is given in the tables (Table 1, Table 2 and Table 3).

Beyond the four groups defined above, patients P1, P4, P6, P7, and P16 were also associated with the loss of the *TSC2* gene, which plays the role of a tumor suppressor. The loss of *TSC2* in 5 of 16 patients deserved our attention and is marked bold in the tables (Table 1, Table 2 and Table 3). Some microdeletions occurred more than once in the entire study population of patients, e.g., *COL1A1* (found in P6 and P7), *NOTCH1* and *MIR4673* (found in P4 and P16) and *GNAS* (found in P4 and P7).

### 2.1. The First Group with MALAT1 and TALAM1 Gain

This group (P1, P2, P3, P5, and P16) was characterized by the gain on chromosome 11 of loci encoding region corresponding to *MALAT1* and *TALAM1* lncRNAs and, in one case (P16), also *MALAT1*-associated small cytoplasmic RNA (MASCRNA). The comparison of gained region encoding *MALAT1*, *TALAM1*, and *MASCRNA* between five patients is displayed in Figure 1. This region was not completely amplified in any patient in this group, so its expression function was questionable. In addition, several aberrations, e.g., giant losses on 6q ($\hat{\alpha }$ = 65%) in P2, 5q ($\hat{\alpha }$ = 59%) in P3, and microdeletion on 16p ($\hat{\alpha }$ = 38%) along with gain on 1p ($\hat{\alpha }$ = 39%) in P1, were also detected in adenoma tissues compared to the adjacent tissues, as shown in Table 1 for P1, P2, and P3. Patients P5 and P16 were described in detail in the separate section depicting the third group of patients with whom they shared common characteristics. This group included both male (P2 and P16) and female (P1, P3 and P5) patients ranging in age from 43 years to 61 years. The adenomas in this group were both tubular (P1, P3 and P5) and tubulo-villous (P2 and P16) in nature with low-grade dysplasia (P1, P3, P5 and P16) and one case of high-grade dysplasia (P2). They were ranked grade 3 according to the Vienna classification (P1, P3, P5 and P16), with one case of grade 4.1 (P2). Adenomas were from both the colon (P1 and P2) and the rectum (P3, P5 and P16).

### 2.2. The Second Group with Microdeletions

Common features of patients P6, P7 and P10 in this second group were many microdeletions at 7q, 9p, 16p, 17q, and 20q in the genome of adenoma tissue compared to that of adjacent tissue, and likewise without the presence of *MALAT1* or *TALAM1* gain on chromosome 11 or any other gain. The list of losses in the adenoma genome of patients is shown in Table 2. Similar aberrations were found in patients P6 and P7, such as loss at 16p of *TSC2* and loss at 17q of *COL1A1*. In other losses, patients differed from each other. Adenomas from these patients were all tubular in nature with low grade dysplasia, ranked by grade 3 according to the Vienna classification and originated from both the colon (P6, P7) and the rectum (P10). The age of the patients ranged from 63 to 68 years, and all were male.

### 2.3. The Third Group with Affected Karyotype

Three patients from the analyzed set showed significant disruption of the adenoma tissue karyotype compared to that of adjacent tissue (Figure 2). There were two females (P4 and P5) and one male (P16) in this group. Rather young age was the common feature shared by the patients: 29 years (P4), 43 years (P5), 43 years (P16). Histologically, they had tubular (P4, P5) to tubulo-villous (P16) adenomas with low grade dysplasia, according to the Vienna classification with grade 3, originating from both the colon (P4) and the rectum (P5 and P16). Since these are serious changes in karyotype that deserve increased attention, we have described the patients in this group in more detail. A section below is dedicated to each patient, including the family and personal history and the reason for the colonoscopy examination. These patients continue to be monitored after adenoma resection.

#### 2.3.1. Patient No. 4

Patient P4 exhibited short arm gains along with long arm losses, as found on chromosomes 8 ($\hat{\alpha }$ = 23%), 10 ($\hat{\alpha }$ = 19%), 17 ($\hat{\alpha }$ = 20%), and 20 ($\hat{\alpha }$ = 32%) and including breaks in the centromeres of chromosomes 10 and 17. Chromosomes 8 and 20 had breaks outside their centromeres. On chromosome 8, the break was on the short arms. The short arm of chromosome 20 was partially duplicated together with a partial deletion of the long arm of this chromosome. Characteristics found in the genome of the pathological clone (Figure 2A) corresponded to the occurrence of isochromosomes or unbalanced translocation (chr: 8, 10, 17 and 20). In addition to these findings, this patient was found to have monosomy of chromosomes 13 ($\hat{\alpha }$ = 14%) and X ($\hat{\alpha }$ = 12%) and trisomy of chromosomes 14 ($\hat{\alpha }$ = 14%) and 18 ($\hat{\alpha }$ = 23%). Other aberrations such as microdeletion on chromosomes 1p, 4p, 9q, 16p, and gain on chromosomes 6q and 7p are listed in Table 3. Interestingly, deletion of loci bearing *MALAT1* and *TALAM1* ($\hat{\alpha }$ = 100%) also occurred in this patient. The patient had no family history either of CRC or other cancer and underwent colonoscopy due to intestinal discomforts such as diarrhea and flatulence. Furthermore, the patient suffered from cholelithiasis, a disease of the bile ducts. The patient was recommended to have a follow-up colonoscopy three years later.

#### 2.3.2. Patient No. 5

In the adenoma genome of patient P5 (Figure 2B), trisomy of three chromosomes (chr: 7 ($\hat{\alpha }$ = 23%), 13 ($\hat{\alpha }$ = 22%) and X ($\hat{\alpha }$ = 11%)) was found, and at the same time gain of the region encoding for *MALAT1* and *TALAM1* ($\hat{\alpha }$ = 100%) (Table 3). The patient underwent a colonoscopy examination based on previous occurrences of adenomas in the colon and family history, and the patient’s sibling suffered from Crohn’s disease. The patient underwent urea dilatation and cervical conization in the past. The patient was recommended to have a follow-up colonoscopy three years later.

#### 2.3.3. Patient No. 16

Based on these data for P16 patient, we believe the pathological clone originally had three sets of chromosomes and one of them was gradually lost (Figure 2C). This patient was found to have trisomy of chromosomes 3 ($\hat{\alpha }$ = 23%), 5 ($\hat{\alpha }$ = 20%), 6 ($\hat{\alpha }$ = 20%), 7 ($\hat{\alpha }$ = 20%), 8 ($\hat{\alpha }$ = 20%), 12 ($\hat{\alpha }$ = 20%), 13 ($\hat{\alpha }$ = 25%), 15 ($\hat{\alpha }$ = 26%), 19 ($\hat{\alpha }$ = 30%), 20 ($\hat{\alpha }$ = 32%), 21 ($\hat{\alpha }$ = 20%), X ($\hat{\alpha }$ = 11%), and Y ($\hat{\alpha }$ = 10%), and one monosomy of chromosome 18 ($\hat{\alpha }$ = 31%). The gain of *MALAT1* and *TALAM1* ($\hat{\alpha }$ = 100%) and many microdeletions on chromosomes 1q, 1p, 9q, 16p, 16q, 17q, and 22q were also found in this sample (Table 3). The patient underwent a planned colonoscopy and was only treated for hemorrhoids. This patient had a family history of CRC and was recommended to have a follow-up colonoscopy three years later.

### 2.4. The Fourth Group with a Negative Finding

The adenoma tissues of patients P8, P9, P11, P12, P13, P14, and P15 showed no changes at the chromosomal level compared to adjacent tissue. No relevant link between patients was found. The age of the group ranged from 44 to 67 years. This group included two women and five men. From a histological point of view, adenomas showed a tubular to tubulo-villous character with low-grade dysplasia (except for P9, who showed high-grade dysplasia), and tissues were taken from both the colon (P8, P12, P13, P14, and P15) and the rectum (P9, P11).

## 3. Discussion

Examination of CIN in precancerous stages is crucial to understand the development of colorectal adenoma. While most studies focus their investigation of CIN in carcinomas, this study took a step back and searched for possible causes of cancer already in adenomas. Carcinoma evolution from colorectal adenoma has been reported in relation with 8p, 17p, 15q, 18p and 18q losses and 5q, 7p, 7q, 8q, 13q, 20p and 20q gains [18,19]. In our study, we observed losses across the whole genome (1p, 1q, 4p, 5q, 6q, 7q, 9p, 9q, 11q, 16p, 16q, 17q, 20q, 22q), whereas gains were pronounced less frequently (1p, 6q, 7p, 11q). In this study, the most significant aberrations were the gain of *MALAT1* and *TALAM1* lncRNAs (found in P1, P2, P3, P5, and P16) and losses of genes such as *TSC2* (in P1, P4, P6, P7 and P16 individuals), *COL1A1* (P6 and P7), *NOTCH1* and *MIR4673* (in P4 and P16) and *GNAS* (in P4 and P7). The genes described above either play a role in signaling pathways (*NOTCH1*), affect transcription (*GNAS*, *MIR4673*, *MALAT1* and *TALAM1*), or are involved in cell structure (*COL1A1*) and proliferation (*TSC2*). Their importance in the cancer process has been previously evidenced and, therefore, their gain or loss in adenoma tissue deserves further attention [20,21,22,23,24,25,26,27,28,29,30,31,32,33,34].

When analyzing the data, we encountered a substantial variety of chromosomal aberrations across all samples. For better orientation in the data, we divided the patients into four groups based on similar features observed in the results. The first group of patients was characterized by one major loss in the genome along with the gain in the region encoding *MALAT1* and *TALAM1* lncRNAs. The physiological functions of MALAT1, referred to also as NEAT2 (nuclear-enriched abundant transcript 2), are in alternative splicing, transcriptional and post-transcriptional regulation, synapse formation, and myogenesis [33]. MALAT1, as an epigenetic player, has been often described in connection with cancer progression [34] or as an inflammatory regulator in diabetic retinopathy [35]. The increased presence of this transcript in cancer tissues compared to non-malignant tissues has been reported not only in lung cancer [36], but also in breast [37], bladder [38], cervical [39], and liver cancer [40], and CRC [40,41,42,43], and was associated with poor OS [44]. MALAT1 has been found to promote cell proliferation and migration of cancer cells by regulating the expression of genes promoting metastases, e.g., *RASSF6*, *HNF4G*, *CA2*, *ROBO1*, *MIA2* [36]. 

LncRNA MALAT1 acts primarily as an epigenetic modulator through small endogenous non-coding RNAs (miRNAs) that are no more than 22 nucleotides in size and control translation and post-transcriptional modifications. MiRNAs have been widely described in the process of carcinogenesis as negatively regulating gene expression in the target gene [45]. The role of the lncRNA MALAT1 in carcinogenesis is through interaction with miRNAs via a “sponge” event, a process whereby competing endogenous RNAs (ceRNAs) share recognition elements (MREs) with miRNAs and thus influence each other’s function [46]. For example, MALAT1 overexpression reduces the expression of miR-145, which under normal conditions inhibits *SOX9*, the gene responsible for differentiation and skeleton development. The inhibition pathway MALAT1/miR-145/*SOX9* thus promotes colorectal cancer cell growth, migration, and invasion [47]. Among other miRNAs, that are target of MALAT1 in relation to colorectal cancer progression, are also miR-508–5p, miR-324-3p, miR-363–3p, and miR-129-5 [48]. The role of MALAT1 in inflammation and cancer progression was demonstrated by Huang et al. in hepatocellular carcinoma, where MALAT1 promotes cancer cell growth by binding Brahma-related gene 1 (BRG1) and recruiting it into the promoter region of *IL-6* and *CXCL8*, thus enabling transcription factors to start the expression of these pro-inflammatory mediators [48]. In addition, Qing et al. discovered that CRC patients with lower expression of *MALAT1* in primary tumors had a better prognosis [44,49]. Another study confirmed increased levels of *MALAT1* in colorectal adenomas and a significant difference between the type and number of polyps compared to unaffected colon epithelium [50]. Therefore, we concluded that amplification of the region encoding *MALAT1* and *TALAM1* in 5 out of the 16 adenoma samples revealed cancer potential in these samples. However, as illustrated in Figure 1, this region is not completely amplified in all patients. The question, therefore, remains whether the resulting product is fully functional. In the first group, in addition to the gain of *MALAT1* and *TALAM1*, losses on chromosomes 5, 6, and 16 were detected. The regions where the losses occurred contained genes associated with CRC: oncogenes (*ROS1*, *FER*) [51,52,53] and tumor suppressors (*APC*, *MCC*, *LOX*) [54,55,56]. The losses may also have contributed to the onset or development of the adenoma.

Microdeletions at 7q, 9p, 16p, 17q, and 20q together with no gains were observed in the intestinal adenoma genome of the second group of patients (P6, P7 and P10). Similar changes in the genome have been observed in other publications studying CIN in colorectal adenomas. Hirsch et al. applied aCGH to 13 FFPE-derived colorectal adenomas rising in high-grade adenomas and revealed losses at 1p, 1, 5q, 8p, 10q, 11q, 16p, 17p, 18q, 18, and 20p, and gains at 4q, 6, 7, 8q, 12p, 12q, 14q, 12, 13, 19, 20q, 20, and X. The study was compromised by high DNA fragmentation due to paraffin fixation [57]. Degraded DNA (length of fragment < 1000 bp) results in biased labeling and inaccurate results. An earlier study from 2002 by Hermsen et al. reported the most frequent 8p, 15q, 17p, and 18q losses and 8q, 13q, and 20q gains in FFPE-derived samples of 66 non-progressed adenomas, 46 progressed adenomas, and 36 colorectal adenomas using chromosomal CGH. However, the DNA extracted from the paraffin was partially degraded, which could have affected the results. The authors of this study observed predominantly losses of chromosomal regions in small non-progressed adenomas, while progressed adenomas were characterized by an increased incidence of gains in the genome [58]. Although the specific losses in our data did not exactly correlate with the already published results, we lean towards the theory that the dominance of microdeletions over gains in genome occurs mainly in non-progressed adenomas, as observed in our samples consisting of tubular, low-grade adenomas without signs of malignancy. The advantage of our study was that we used fresh frozen samples instead of FFPE samples where DNA is often highly degraded. At the same time, we applied the most modern approach to the current CGH using high-resolution array CGH. Thanks to these modifications, we obtained more accurate and reliable results.

The third group included patients with a severely disrupted karyotype. In patient P4 abnormal formations of chromosomes 8, 10, 17 and 20 were found. We assume these could be either isochromosomes or unbalanced translocations. The gain of short arms and loss of long arms, which we could see at chromosomes 10 and 17, suggested that these could be isochromosomes. The occurrence of isochromosomes in colorectal adenomas has been reported only in cell lines derived from familial polyposis carcinoma on chromosomes 1, 14 and translocation on chromosomes 17 and X [59]. In patient P5, trisomy of chromosomes 7, 13, and X occurred in adenoma tissue. A similar result was obtained by Longy et al., who analyzed 25 colonic adenomatous polyps from patients with average age of 66 years, and found the most frequent trisomy on chromosomes 7 (in eight cases) and 13 (in seven cases) using a direct method of chromosome visualization as in prenatal analysis [60]. Another study involving 20 colorectal non-progressed adenomas described the most common gains on chromosomes 3, 7, 13, and 20 and the most common structural rearrangements on chromosomes 1, 13, 17, and 18 using chromosome banding analysis for direct chromosome visualization [61]. Chromosome trisomy in adenoma tissue was a common feature in patient P16 (chr: 3, 5, 6, 7, 8, 12, 13, 15, 19, 20, 21, X, and Y), suggesting that the cells of pathological clone in adenoma tissue had a triploid set of chromosomes, and one of them was gradually lost. Despite the scarcity of studies published on CIN in colorectal adenomas, we hypothesize that chromosome trisomy in this aberrant kind of tissue is a common feature. The only link between all three patients with disrupted karyotypes was their young age at the time of adenoma resection compared to the rest of the study set. We assume that young age is one of the factors contributing to such extensive karyotype diversity in cells of adenoma tissue. Another hypothesis could be that the occurrence of adenomas at such a young age is precisely due to a large-scale change in the genome. Due to the finding of adenomas at such a young age, patients will continue to be monitored and have been advised to have a more frequent colonoscopy examination. Over the last 40 years, the incidence and mortality of CRC have increased in individuals under the age of 50, especially those aged 40–44. In the United States, 11% of colon cancer cases and 18% of rectal cancer cases affect patients under the age of 50. In addition to hereditary CRC syndromes, which appear at an early age (20–30 years), the main causes are sedentary lifestyle, obesity, and associated diabetes mellitus [62,63]. Therefore, it would be appropriate to monitor chromosomal rearrangements in adenoma samples of such young people (<50 years) and to validate the aberrations found in a wider group of patients. In this study, our candidates would be mainly the gain of *MALAT1* and *TALAM1* and the loss of *TSC2*.

From another point of view, we must consider the loss of *TSC2*, which occurred in five patients (P1, P4, P6, P7 and P16) out of 16. Tuberin, a product of *TSC2*, interacts with the hamartin protein, a product of the *TSC1*, in the cell. The main function of these proteins is to activate GTPase proteins, cyclins and many other proteins participating in cell cycle regulation to control the growth and size of cells, thus playing the role of tumor suppressors [20]. Disruption of the signaling pathway moderated by TSC1/TSC2 inhibition complex is often found in cancer development; in particular, loss of TSC2 function leads to hyperactivation of the mechanistic target of rapamycin complex 1 (mTORC1), a protein complex responsible for activation of protein translation [21,64]. Increased de novo protein translation enhanced endoplasmic reticulum (ER) stress, a common sign in cancer cells [65]. *TSC1* and *TSC2* are also potent regulators of the expression of a transmembrane protein named Programmed cell death ligand 1 (PD-L1), a target of inhibitors in non–small cell lung cancer treatment. The deficiency of *TSC2* showed up-regulation of PD-L1 in human lung cancer cell lines [66]. *TSC1* and *TSC2* are growth suppressor genes, therefore, we conclude that loss of *TSC2* could contribute to adenoma development.

In seven patients from the entire study group, no change was found at the chromosomal level in adenoma tissues compared to adjacent tissues. The negative finding can be justified by the fact that CIN does not occur in all colorectal tumors, but only in 70% of cases [67], thus making the expectation of CIN occurring in colorectal adenomas even lower. Furthermore, using aCGH, we were unable to detect driver mutations in genes that may be responsible for the development of adenoma. The negative findings could also be caused by insufficient genome coverage in the applied array. We could not detect aberrations in mosaicism lower than 15% certainty. Another factor was that aCGH did not capture balanced rearrangements that can disrupt genes and gene architecture or affect position effect expression at break sites.

The hypothesis that CIN will already appear early in adenoma tissue has been confirmed. Nine samples out of 16 showed CIN, representing 56% of the total set, while in CRC, CIN occurred in 65–70% [12]. We assumed that CIN would be slightly less represented in adenomas than in CRC, which was also confirmed. The histological classification of the adenomas according to clinical experience did not correlate with the extent of CIN among all samples of adenomas. Samples from patients were intentionally selected with the closest possible histological similarity to find their common feature also at the chromosomal level. The results showed a great diversity of CIN across the whole study group. An important finding was that the disrupted karyotype in adenoma tissue cells in the third group of patients (P4, P5 and P16) was also associated with young age, which may have contributed to such variable CIN. The aberrations found in the third group should receive more attention and validation with samples from patients under 50 years of age. The aberrations found in this study (losses of *TSC2*, *COL1A1*, *NOTCH1*, *MIR4673* and *GNAS*, and gains of *MALAT1* and *TALAM1*) may serve as candidate biomarkers for early detection of colorectal cancer onset.

The main clinical significance of our findings will be shown as we continue to monitor these patients to observe how/if their health conditions develop differently (e.g., whether the group with large changes will have more adenomas or progress to cancer). Our results raise doubts as to whether the classification based on histology is sufficient and suggest it should be extended to genetic analysis (e.g., detection of chromosomal instability or detection of specific changes—the diagnosis of the *MALAT1*/*TALAM1* region is offered).

Nowadays, it is crucial to identify biomarkers with sufficient sensitivity and specificity that can predict the early transformation of intestinal adenoma into adenocarcinoma. Detection of disease at its origin would help suppress the development into cancer and thus improve the patient’s prognosis. These new biomarkers could serve physicians as indicators for colonoscopy and predict the frequency of this examination.

## 4. Materials and Methods

### 4.1. Sample Collection

The study included 16 individuals (Table 4) with adenomas of tubular or tubulo-villous histology that underwent recommended colonoscopy examination due to prevention or for intestinal discomfort, such as diarrhea or flatulence. The collection of samples was carried out in cooperation with the Department of Gastroenterology at Thomayer Hospital in Prague and with the Clinic of Hepatogastroenterology at the Institute for Clinical and Experimental Medicine, (Prague, Czech Republic) between March 2017 and December 2020. The study was approved by the ethical committees of both institutions. Sample biopsies of adenomas and adjacent tissue were placed into stabilization solution RNA and then stored in a deep-freeze box at −80 °C. Adenomas were transferred to histopathological examination to confirm that the samples did not show any signs of malignancy. 

All subjects included in the study provided written informed consent to participate in the study and to use their biological samples for genetic analyses, in accordance with the Helsinki Declaration. The design of the study was also approved by the Ethics Committee of the Institute of Experimental Medicine, Prague, Czech Republic.

### 4.2. DNA Extraction

The genomic DNA was extracted from disrupted adenomas and adjacent tissues using AllPrep DNA/RNA Mini Kit (Qiagen, Düsseldorf, Germany) according to the standard protocol. MagNa Lyser Green Beads (Roche, Munich, Germany) were used for tissue disruption in a homogenizer (MagNaLyser Instrument, Version 4, Roche, Mannheim, Germany). The concentration of isolated DNA was measured by Qubit™ dsDNA BR Assay Kit (Invitrogen, Waltham, MA, USA) on Qubit 3.0 fluorometer (Invitrogen, Waltham, MA, USA).

### 4.3. Comparative Genomic Hybridization Array Design

Arrays used in the present study were designed to cover genomic regions bearing the cancer-associated genes by SurePrint G3 Cancer CGH+ single nucleotide polymorphism (SNP) Microarray Kit, 4 × 180 K (Agilent, Santa Clara, CA, USA). Instead of commercial DNA included within the kit, DNA isolated from the adjacent tissue from each patient was used as a source of reference DNA. For DNA labeling with Cy5 and Cy3 labels, Sure Tag Complete DNA Labeling enzyme kit was purchased (Agilent, Santa Clara, CA, USA); Cy5 label was applied as a reference for adjacent tissues and Cy3 label for adenoma tissues. The input amount of DNA into the labeling was on average 850 ng. Hybridization was performed by using an Oligo aCGH/ChIP-on-chip Hybridization kit (Agilent, Santa Clara, CA, USA).

### 4.4. Array Processing and Bioinformatics Data Analysis

SureScan Microarray Scanner instrument (Agilent, Santa Clara, CA, USA) with program Agilent G3 CGH corresponding to the barcode of arrays was used for array scanning. Data were processed in Agilent CytoGenomics software with the default analysis method—CGH v2. Due to the use of patient tissues as reference samples, SNP probes could not be analyzed.

The percentage of mosaicism was estimated according to the formula below (1), where δ is the observed fold change of the mean log ratio ($\delta =2^{logR})$. The formula was introduced by Cheung et al. [68] to determine the presence of two or multiple cell lines ($\hat{\alpha }$ ≤ 15% implies the absence or low frequency of mosaic, and $\hat{\alpha }$ > 15% indicates the presence of more than one clone in the cell population of the sample). Formula (1) is designed to calculate mosaicism in autosomes; we have modified the formula for heterochromosomes for male samples (2).
(1)$$\hat{\alpha }=\left|\frac{\delta -1}{0.5}\right|\times 100$$
(2)$$\hat{\alpha }=\left|\frac{\delta -1}{1}\right|\times 100$$

## 5. Conclusions

Using the aCGH method, we analyzed paired samples of colorectal adenomas and adjacent tissue from a total of 16 patients with histologically similar samples. The presence of CIN in the precancerous stages was confirmed in 56% of the adenomas. The significant gain was found on 11q13.1, encoding for *MALAT1* and *TALAM1* lncRNAs in five patients. We further identified several losses on chromosomes 1p, 1q, 4p, 5q, 6q, 7q, 9p, 9q, 11q, 16p, 16q, 17q, 20q, and 22q and gains in the 1p, 6q, 7p, and 11q regions. Overall, losses outweighed gains in adenoma tissue. Losses that were identified in at least two patients included the *TSC2*, *COL1A1*, *NOTCH1*, *MIR4673* and *GNAS* genes. *TSC2* loss was detected in five patients; since it is a tumor suppressor gene, we assume that its absence is involved in adenoma formation. Seven patients did not display any CIN in adenoma tissue at all. The study provided novel insight into chromosomal rearrangements in colorectal adenomas and provided new candidate biomarker lncRNA MALAT1 for further investigation.

## Figures and Tables

**Figure 1 ijms-23-07656-f001:**
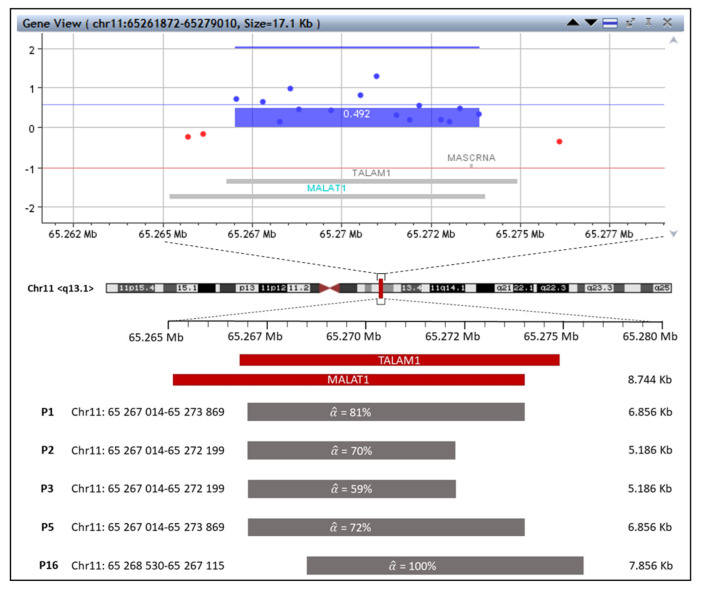
The representative example of gain in region encoding *MALAT1*, *TALAM1*, and *MASCRNA* is pictured as a blue rectangle in the region at chr11, q13.1, in adenoma of P1 at the top of the image. Thin horizontal red and blue lines represent values corresponding to the non-mosaic state of deletions (−1) or duplications (0.58). A comparison of this gained region between patients from the first group (P1, P2, P3, P5, and P16) is shown at the bottom of the image. Mosaicism of each patient is expressed by the symbol $\hat{\alpha }$.

**Figure 2 ijms-23-07656-f002:**
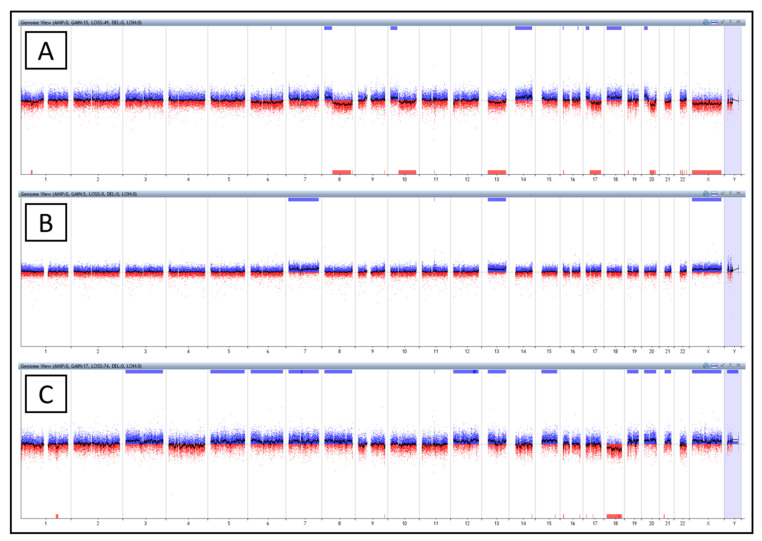
The entire genome of patient P4 (**A**), P5 (**B**), P16 (**C**). (**A**) Trisomy of chromosomes 14 and 18, monosomy of chromosomes 13 and X, aberrations on chromosomes 1, 6, 8, 10, 11, 17, 20 at P4. (**B**) Trisomy of chromosomes 7, 13, X and aberration on chromosome 11 at P5. (**C**) Trisomy of chromosomes 3, 5, 6, 7, 8, 12, 13, 15, 19, 20, 21, X, Y, monosomy of chromosome 18, and aberration on chromosome 11 at P16.

**Table 1 ijms-23-07656-t001:** Additional aberrations found in patients P1, P2 and P3.

Patient	Type	Chromosome	Location	Cytoband	Size	${\hat{\alpha \;}}^{a}$	Gene Name
P1	loss	16	2,110,696–2,136,380	p13.3	25.685 kb	38%	*TSC2*
	gain	1	2,260,756–3,080,894	p36.33–p36.32	820.139 kb	39%	*PEX10,**PLCH2*, *PANK4*, *HES5*, *TNFRSF14*, *MMEL1*, *ACTRT2*, *PRDM16*, *MORN1*, *LOC100129534*, *RER1*, *TNFRSF14-AS1*, *LOC100996583*, *PRXL2B*, *TTC34*, *PRDM16-DT*, *MIR4251*
		11	65,267,014–65,273,869	q13.1	6.856 kb	81%	*MALAT1*, *TALAM1*
P2	loss	6	107,338,10–127,407,686	q21–q22.33	20,069.584 kb	65%	*MTRES1*, *BEND3*, *PDSS2*, *SOBP*, *SEC63*, *OSTM1*, *NR2E1*, *SNX3*, *AFG1L*, *FOXO3*, *ARMC2*, *SESN1*, *CD164*, *SMPD2*, *MICAL1*, *ZBTB24*, *AK9*, *FIG4*, *GPR6*, *WASF1*, *CDC40*, *DDO*, *SLC22A16*, *CDK19*, *AMD1*, *GTF3C6*, *RPF2*, *SLC16A10*, *MFSD4B*, *REV3L*, *TRAF3IP2*, *FYN*, *CCN6*, *TUBE1*, *LAMA4*, *MARCKS*, *HDAC2*, *HS3ST5*, *FRK*, *COL10A1*, *DSE*, *TSPYL1*, *CALHM6*, *TRAPPC3L*, *RSPH4A*, *KPNA5*, *GPRC6A*, *RFX6*, *VGLL2*, *ROS1*, *GOPC*, *NUS1*, *PLN*, *MCM9*, *ASF1A*, *MAN1A1*, *TBC1D32*, *GJA1*, *HSF2*, *SERINC1*, *PKIB*, *FABP7*, *SMPDL3A*, *CLVS2*, *TRDN*, *NKAIN2*, *RNF217-AS1*, *RNF217*, *TPD52L1*, *HEY2*, *NCOA7*, *HINT3*, *CENPW*, *SCML4*, *OSTM1-AS1*, *LINC00222*, *ARMC2-AS1*, *CEP57L1*, *CCDC162P*, *PPIL6*, *METTL24*, *SNORA40C*, *GSTM2P1*, *SNORD166*, *TRAF3IP2-AS1*, *LINC02527*, *FAM229B*, *LOC101927640*, *RFPL4B*, *LINC02518*, *LINC02541*, *MROCKI*, *FLJ34503*, *HDAC2-AS2*, *LINC02534*, *TPI1P3*, *NT5DC1*, *TSPYL4*, *LOC100287467*, *CALHM5*, *CALHM4*, *RWDD1*, *ZUP1*, *FAM162B*, *DCBLD1*, *LOC101927919*, *SLC35F1*, *LOC105377967*, *CEP85L*, *BRD7P3*, *SELENOKP3*, *FAM184A*, *MIR548B*, *LOC285762*, *LOC105377975*, *MIR3144*, *TRDN-AS1*, *LOC100126584*, *HDDC2*, *LINC02523*, *NCOA7-AS1*, *TRMT11*, *MIR588*
	gain	11	65,267,014–65,272,199	q13.1	5.186 kb	70%	*MALAT1, TALAM1*
P3	loss	5	102,026,08–127,375,136	q21.1–q23.3	25,349.057 kb	59%	*PAM*, *PPIP5K2*, *C5orf30*, *NUDT12*, *EFNA5*, *FBXL17*, *FER*, *MAN2A1*, *SLC25A46*, *TSLP*, *WDR36*, *CAMK4*, *STARD4*, *NREP*, *EPB41L4A*, *APC*, *SRP19*, *REEP5*, *DCP2*, *MCC*, *TSSK1B*, *YTHDC2*, *KCNN2*, *TRIM36*, *PGGT1B*, *FEM1C*, *TICAM2*, *CDO1*, *ATG12*, *AP3S1*, *LVRN*, *COMMD10*, *SEMA6A*, *DMXL1*, *TNFAIP8*, *HSD17B4*, *FAM170A*, *PRR16*, *FTMT*, *SRFBP1*, *LOX*, *SNCAIP*, *SNX2*, *PPIC*, *PRDM6*, *CEP120*, *CSNK1G3*, *ALDH7A1*, *PHAX*, *LMNB1*, *MARCHF3*, *MEGF10*, *GIN1*, *LINC02115*, *RAB9BP1*, *LINC01950*, *LINC01023*, *LOC285638*, *PJA2*, *LINC01848*, *TMEM232*, *MIR548F3*, *STARD4-AS1*, *NREP-AS1*, *EPB41L4A-AS1*, *SNORA13*, *LOC101927023*, *EPB41L4A-DT*, *LINC02200*, *LOC102467216*, *LOC101927078*, *LINC01957*, *CCDC112*, *TMED7-TICAM2*, *LOC101927100*, *TMED7*, *LOC102467217*, *LINCADL*, *ARL14EPL*, *MIR12130*, *LOC101927190*, *SEMA6A-AS1*, *SEMA6A-AS2*, *LINC02214*, *LINC00992*, *LINC02147*, *LINC02148*, *LINC02208*, *LINC02215*, *DTWD2*, *MIR1244-1*, *MIR1244-2*, *MIR1244-3*, *MIR1244-4*, *LOC105379143*, *MIR5706*, *LOC102467226*, *ZNF474*, *LOC100505841*, *MGC32805*, *LOC101927357*, *LINC02201*, *SNX24*, *LOC105379152*, *LINC01170*, *ZNF608*, *LOC101927421*, *LINC02240*, *LINC02039*, *LOC101927488*, *GRAMD2B*, *TEX43*, *LMNB1-DT*, *C5orf63*, *PRRC1*, *CTXN3*, *CCDC192*, *LINC01184*
	gain	11	65,267,014–65,272,199	q13.1	5.186 kb	70%	*MALAT1*, *TALAM1*

^a^ percentage of mosaicism.

**Table 2 ijms-23-07656-t002:** List of losses in adenoma tissues of patients P6, P7, and P10 (second group).

Patient	Type	Chromosome	Location	Cytoband	Size	${\hat{\alpha }}^{a}$	Gene Name
P6	loss	16	2,103,321–2,138,073	p13.3	34.753 kb	51%	*TSC2*
		17	48,263,792–48,273,321	q21.33	9.530 kb	34%	*COL1A1*
P7	loss	7	73,442,449–73,481,111	q11.23	38.663 kb	35%	*ELN*
		16	2,105,434–2,138,073	p13.3	32.640 kb	58%	*TSC2*
		17	48,263,792–48,273,777	q21.33	9.986 kb	40%	*COL1A1*
		20	57,407,840–57,495,925	q13.32	88.086 kb	38%	*GNAS-AS1*, *GNAS*, *LOC1019*
P10	loss	9	21,549,338–23,792,459	p21.3	2243.122 kb	46%	*MIR31HG*, *MTAP*, *CDKN2A*, *CDKN2B-AS1*, *CDKN2B*, *DMRTA1*, *ELAVL2*, *CDKN2A-DT*, *LINC01239*, *LOC101929563*

^a^ percentage of mosaicism.

**Table 3 ijms-23-07656-t003:** List of aberrations found in P4, P5, and P16 (third group).

Patient	Type	Chromosome	Location	Cytoband	Size	${\hat{\alpha }}^{a}$	Gene Name
P4	loss	1	55,108,604–61,921,519	p32.3–p31.3	6812.92 kb	33%	*TTC4*, *PARS2*, *LEXM*, *DHCR24*, *BSND*, *PCSK9*, *USP24*, *PLPP3*, *PRKAA2*, *FYB2*, *C8A*, *C8B*, *DAB1*, *OMA1*, *TACSTD2*, *MYSM1*, *JUN*, *FGGY*, *HOOK1*, *CYP2J2*, *NFIA*, *MROH7*, *MROH7-TTC4*, *TTC22*, *TMEM61*, *LOC100507634*, *MIR4422HG*, *MIR4422*, *LINC01753*, *LINC01755*, *LINC01767*, *LOC101929935*, *DAB1-AS1*, *LINC01135*, *LINC01358*, *HSD52*, *MIR4711*, *LOC101926944*, *C1orf87*, *LINC01748*, *LOC101926964*, *NFIA-AS2*, *NFIA-AS1*
		4	1,802,707–1,809,469	p16.3	6.763 kb	65%	*FGFR3*
		9	139,394,991–139,418,283	q34.3	23.293 kb	50%	*NOTCH1*, *MIR4673*
		11	65,265,673–65,273,325	q13.1	7.653 kb	100%	*MALAT1*, *TALAM1*
		16	2,105,434–2,138,073	p13.3	32.640 kb	50%	*TSC2*
	gain	6	107,068,675–109,166,111	q21	2097.437 kb	47%	*RTN4IP1*, *QRSL1*, *MTRES1*, *BEND3*, *PDSS2*, *SOBP*, *SEC63*, *OSTM1*, *NR2E1*, *SNX3*, *AFG1L*, *FOXO3*, *LINC02526*, *LINC02532*, *MIR587*, *SCML4*, *OSTM1-AS1*, *LINC00222*
		7	83,325–2,737,748	p22.3	2654.42 kb	40%	*FAM20C*, *PDGFA*, *PRKAR1B*, *DNAAF5*, *SUN1*, *GET4*, *ADAP1*, *COX19*, *CYP2W1*, *MIR339*, *GPER1*, *ZFAND2A*, *INTS1*, *MAFK*, *PSMG3*, *ELFN1*, *MAD1L1*, *MRM2*, *NUDT1*, *SNX8*, *EIF3B*, *CHST12*, *LFNG*, *BRAT1*, *IQCE*, *TTYH3*, *AMZ1*, *LOC102723672*, *LOC100507642*, *LOC105375115*, *LOC442497*, *HRAT92*, *LOC101927000*, *LOC101926963*, *C7orf50*, *GPR146*, *LOC101927021*, *UNCX*, *MICALL2*, *LOC100128653*, *TMEM184A*, *PSMG3-AS1*, *TFAMP1*, *ELFN1-AS1*, *MIR4655*, *SNORA114*, *MIR6836*, *GRIFIN*, *MIR4648*
P4	monosomy	8q		-		25%	-
		13				14%	
		17q				20%	
		20q				31%	
		X				12%	
	trisomy	8p		-		23%	-
		10p				19%	
		14				23%	
		17p				20%	
		18				23%	
		20p				31%	
P5	gain	11	65,267,014–65,273,869	q13.1	6.856 kb	100%	*MALAT1*, *TALAM1*
	trisomy	7		-		23%	-
		13				22%	
		X				11%	
P16	loss	1	185,274268–199,118,773	q25.3–q32.1	13,828 kb	24%	*HMCN1*, *PRG4*, *TPR*, *ODR4*, *OCLM*, *PDC*, *PTGS2*, *PACERR*, *PLA2G4A*, *BRINP3*, *RGS18*, *RGS21*, *RGS1*, *RGS13*, *RGS2*, *UCHL5*, *RO60*, *GLRX2*, *CDC73*, *B3GALT2*, *KCNT2*, *CFH*, *CFHR3*, *CFHR1*, *CFHR4*, *CFHR2*, *CFHR5*, *F13B*, *ASPM*, *CRB1*, *DENND1B*, *LHX9*, *NEK7*, *ATP6V1G3*, *PTPRC*, *MIR181B1*, *MIR181A1*, *LOC102724919*, *LINC01036*, *LINC01037*, *LINC01351*, *LINC01720*, *LINC01680*, *MIR4426*, *LINC01032*, *SCARNA18B*, *MIR1278*, *LINC01031*, *LINC01724*, *MIR4735*, *ZBTB41*, *C1orf53*, *MIR181A1HG*, *LINC01222*, *LINC01221*, *LINC02789*
		1	890,945–3,729,090	p36.33–p36.32	2838.146 kb	40%	*NOC2L*, *PERM1*, *HES4*, *ISG15*, *AGRN*, *MIR200B*, *MIR200A*, *MIR429*, *TNFRSF18*, *TNFRSF4*, *SDF4*, *B3GALT6*, *C1QTNF12*, *SCNN1D*, *INTS11*, *CPTP*, *TAS1R3*, *DVL1*, *MXRA8*, *AURKAIP1*, *CCNL2*, *MRPL20*, *VWA1*, *ATAD3C*, *ATAD3B*, *ATAD3A*, *TMEM240*, *SSU72*, *MIB2*, *MMP23B*, *MMP23A*, *CDK11B*, *CDK11A*, *NADK*, *GNB1*, *CALML6*, *GABRD*, *PRKCZ*, *FAAP20*, *SKI*, *PEX10*, *PLCH2*, *PANK4*, *HES5*, *TNFRSF14*, *MMEL1*, *ACTRT2*, *PRDM16*, *MEGF6*, *MIR551A*, *TPRG1L*, *WRAP73*, *TP73*, *SMIM1*, *CEP104*, *KLHL17*, *PLEKHN1*, *LOC100288175*, *RNF223*, *C1orf159*, *LINC01342*, *TTLL10*, *UBE2J2*, *ACAP3*, *MIR6726*, *SNORD167*, *PUSL1*, *MIR6727*, *MIR6808*, *MRPL20-AS1*, *ANKRD65*, *TMEM88B*, *LINC01770*, *FNDC10*, *LOC105378586*, *SLC35E2B*, *SLC35E2A*, *TMEM52*, *CFAP74*, *LOC105378591*, *PRKCZ-AS1*, *MORN1*, *LOC100129534*, *RER1*, *TNFRSF14-AS1*, *LOC100996583*, *PRXL2B*, *TTC34*, *PRDM16-DT*, *MIR4251*, *ARHGEF16*, *TP73-AS1*, *CCDC27*, *LRRC47*
P16	loss	9	139,389,744–139,440,753	q34.3	51.010 kb	70%	*NOTCH1*, *MIR4673*, *MIR4674*
		16	2,044,093–2,263,638	p13.3	219.546 kb	52%	*SYNGR3*, *ZNF598*, *NPW*, *SLC9A3R2*, *NTHL1*, *TSC2*, *PKD1*, *MIR1225*, *RAB26*, *TRAF7*, *CASKIN1*, *MLST8*, *PGP*, *LOC105371049*, *MIR6511B1*, *MIR6511B2*, *MIR4516*, *MIR3180-5*, *SNHG19*, *SNORD60*, *BRICD5*
		16	88,365,786–89,383,369	q24.2–q24.3	1017.584 kb	43%	*ZNF469*, *ZFPM1*, *IL17C*, *CYBA*, *MVD*, *SNAI3*, *RNF166*, *CTU2*, *PIEZO1*, *CDT1*, *APRT*, *GALNS*, *TRAPPC2L*, *CBFA2T3*, *ACSF3*, *CDH15*, *ANKRD11*, *MIR5189*, *LOC100128882*, *ZC3H18-AS1*, *ZC3H18*, *SNAI3-AS1*, *MIR4722*, *LOC100289580*, *LOC339059*, *PABPN1L*, *LOC101927793*, *LOC100129697*, *LINC00304*, *LINC02138*, *SLC22A31*, *ZNF778*, *LOC105371414*
	loss	17	36,861,875–36,896,355	q12	34.481 kb	54%	*MLLT6*, *CISD3*, *PCGF2*, *MIR4726*
		22	19,702,774–19,851,138	q11.21	148.365 kb	47%	*SEPTIN5*, *SEPT5-GP1BB*, *GP1BB*, *TBX1*, *GNB1L*, *RTL10*
	gain	11	65,268,530–65,276,115	q13.1	7.586 kb	100%	*MALAT1*, *TALAM1*, *MASCRNA*
	monosomy	18		-		31%	-
	trisomy	3		-		23%	-
		5				20%	
		6				20%	
		7				20%	
		8				20%	
		12				20%	
		13				25%	
		15				26%	
		19				30%	
		20				32%	
		21				20%	
		X				11%	
		Y				10%	

^a^ percentage of mosaicism.

**Table 4 ijms-23-07656-t004:** Clinical and histological characteristics. ^a^ Female, ^b^ Male, ^c^ Tubular, ^d^ Tubulo-villous, ^e^ Low grade, ^f^ High grade. ^g^ For small samples, only two dimensions were given by pathologists.

Sample ID	Gender	Age	Histology Type of Adenoma	Size of Adenoma (mm)	Vienna Classification	Grade	Localization
P1	F ^a^	61	T ^c^	15 × 8 × 10	3	LG ^e^	colon
P2	M ^b^	60	TL ^d^	7 × 4 × 4	4.1.	HG ^f^	colon
P3	F	56	T	24 × 12 × 8	3	LG	rectum
P4	F	29	T	8 × 8 ^g^	3	LG	colon
P5	F	43	T	7 × 5 ^g^	3	LG	rectum
P6	M	64	T	4 × 2 ^g^	3	LG	colon
P7	M	63	T	3 × 4 ^g^	3	LG	colon
P8	M	67	T	4 × 2 × 2	3	LG	colon
P9	M	61	T	5 × 5 ^g^	4.1.	HG	rectum
P10	M	68	T	12 × 8 × 10	3	LG	rectum
P11	F	44	TL	10 × 10 × 4	3	LG	rectum
P12	M	53	T	3 × 10 ^g^	3	LG	colon
P13	M	54	T	9 × 3 × 3	3	LG	colon
P14	M	49	TL	10 × 3 × 2	3	LG	colon
P15	F	57	T	2 × 2 ^g^	3	LG	colon
P16	M	43	TL	18 × 13 × 11	3	LG	rectum

## Data Availability

The data presented in this study are available on request from the corresponding author. The data are not publicly available due to privacy and ethical restrictions.

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
