# Peer review of "Discovery of Long Non-Coding RNA MALAT1 Amplification in Precancerous Colorectal Lesions"

_ijms, 2022, doi:10.3390/ijms23147656_

Round 1

Reviewer 1 Report

l   It is suggested to discuss more about the possible mechanism underlying the contribution of MALAT1 in the malignant transformation from the literature review. For instance, it has been shown that colonic adenoma cells may transform into adenocarcinoma cells through inflammation. Aside from the promotion of migration and invasion, it may be better to include more information regarding the potential role of MALAT1 in cancer formation.

l   Although studies about the loss of TSC2 in colorectal cancer are limited, it is still worthy to mention more about the significance/ function of TSC in tumor development, such as the signaling pathways that are affected.

l   Please check the grammar carefully. For instance, the first sentence in the discussion section needs to be corrected. 

Author Response

Reply to reviewer 1

I   It is suggested to discuss more about the possible mechanism underlying the contribution of MALAT1 in the malignant transformation from the literature review. For instance, it has been shown that colonic adenoma cells may transform into adenocarcinoma cells through inflammation. Aside from the promotion of migration and invasion, it may be better to include more information regarding the potential role of MALAT1 in cancer formation.

We thank Reviewer for her/his comment. We have added to the discussion a more detailed description of the role of MALAT1 lncRNA in cancer transition, especially its interaction with miRNA.

MALAT1 has been found to promote cell proliferation and migration of cancer cells by regulating the expression of genes promoting metastases e.g., RASSF6, HNF4G, CA2, ROBO1, MIA2 (doi.org/10.1158/0008-5472.Can-12-2850).

LncRNA MALAT1 acts primarily as an epigenetic modulator through small endogenous non-coding RNAs (miRNAs) that are no more than 22 nucleotides in size and control translation and post-transcriptional modifications. MiRNA have been widely described in the process of carcinogenesis by negatively regulating gene expression in the target gene (doi.org/10.1038/sigtrans.2015.4). The role of the lncRNA MALAT1 in carcinogenesis is through interaction with miRNAs via a "sponge" event, a process whereby competing endogenous RNAs (ceRNAs) share recognition elements (MREs) with miRNAs and thus influence each other's function (doi.org/10.1016/j.arbr.2019.03.018). For example, MALAT1 overexpression reduces the expression of miR-145, which under normal conditions inhibits SOX9, the gene responsible for differentiation and skeleton development. The inhibition pathway MALAT1/miR-145/SOX9 thus promotes colorectal cancer cell growth, migration, and invasion (doi.org/10.1186/s10020-018-0050-5). Among other miRNAs, which are targeted for MALAT1 in relation to colorectal cancer progression, are miR-508–5p, miR-324-3p, miR-363–3p, and miR-129-5 (doi.org/10.1016/j.biopha.2021.111389). MALAT1 role in inflammation cancer progression was demonstrated by Huang et al. in hepatocellular carcinoma, where MALAT1 promotes cancer cell growth by binding Brahma-related gene 1 (BRG1) and recruiting it into the promoter region of IL-6 and CXCL8 and thus enabled transcription factors to start the expression of these pro-inflammatory mediators (doi.org/10.1080/2162402X.2018.1518628).

II   Although studies about the loss of TSC2 in colorectal cancer are limited, it is still worthy to mention more about the significance/ function of TSC in tumor development, such as the signaling pathways that are affected.

          We thank the Reviewer for her/his comment. We agree that the TSC2 function deserves more description in our MS. Therefore, we included in the discussion a functional description of TSC2 in a signaling pathway.

Disruption of signaling pathway moderated by TSC1/TSC2 inhibition complex is often found in cancer development, especially loss of TSC2 function leads to hyperactivation of the mechanistic target of rapamycin complex 1 (mTORC1), a protein complex responsible for activation of protein translation (doi:10.1093/carcin/bgq142, doi.org/10.1016/j.molmed.2010.05.001). Increased de novo protein translation enhanced endoplasmic reticulum (ER) stress, a common sign in cancer cells (doi.org/10.1038/s41388-018-0381-2). TSC1 and TSC2 are also potent regulators of the expression of a transmembrane protein named Programmed cell death ligand 1 (PD-L1), a target of inhibitors in non–small cell lung cancer treatment. The deficiency of TSC2 showed up-regulation of PD-L1 in human lung cancer cell lines (doi.org/10.1126/sciadv.abi9533). TSC1 and TSC2 are growth suppressor genes, therefore, we conclude that loss of TSC2 could contribute to adenoma development.

III   Please check the grammar carefully. For instance, the first sentence in the discussion section needs to be corrected. 

We thank the Reviewer for suggesting the MS grammar. We admit that the first sentence in the discussion was too long, and the meaning may have been obscured. Thus, we have rewritten the sentence into the following two sentences.

Examination of CIN in precancerous stages is crucial to understanding the development of colorectal adenoma. While most studies focus their investigation of CIN in carcinomas, this study takes a step back and looks for possible causes of cancer already in adenomas.

Reviewer 2 Report

The authors use comparative genomic hybridization arrays to study CIN in colorectal adenomas. The methods used are appropriate and the results are clearly described. Early-stage colon carcinogenesis is a valuable area for study. They report amplification of MALAT1, and this would be an interesting observation. The conclusions drawn are not always consistent with the data.

General issues

(1)   The sample size is small (n= 16), with the datapoints being divided into four subclasses. This means that there are only a small number of cases for each, making it difficult to draw general conclusions.

(2)   The authors state that the MALAT1 region is “not completely amplified in any patient…..so its expression function is questionable.” This is well shown in Fig 1.

(3)   It is unclear as to whether patients were screened for MMR gene mutations and accessed for hypermutator phenotype.  

(4)   It would be useful to include a measure of polyp size.

Further experiments could strengthen the MS

(1)   The question of MALAT1 function would be approached by qRT-PCR and/or in situ hybridisation to demonstrate MALAT1 expression is increased compared to normal colon mucosa.

(2)   DNA sequencing would demonstrate MALAT1 amplification within the five group 1 polyps.

Summary

The demonstration of amplified MALAT1 in polyps would be an interesting observation. The current MS is incomplete. Further investigation strategies might increase the identification of genetic defects within the polyps tested and may provide more complete data to support these conclusions.

Author Response

Reply to reviewer 2 

The authors use comparative genomic hybridization arrays to study CIN in colorectal adenomas. The methods used are appropriate and the results are clearly described. Early-stage colon carcinogenesis is a valuable area for study. They report amplification of MALAT1, and this would be an interesting observation. The conclusions drawn are not always consistent with the data.

We thank the Reviewer for his/her comment, based on which we added our findings from the data to the conclusion.

TSC2 loss was detected in five patients; since it is a tumor suppressor gene, we assume that its absence is involved in adenoma formation.

General issues

(1)   The sample size is small (n= 16), with the datapoints being divided into four subclasses. This means that there are only a small number of cases for each, making it difficult to draw general conclusions.

We are thankful for the note. Initially, we intended to have one big uniform group based on a similar histological classification. However, aCGH results showed that the same histology character does not always reflect also same adenomas’ molecular background Therefore, we decided to stratify patients' results according to the similar characteristics (gains in regions encoding long non-coding RNA MALAT1, microdeletions, disrupted karyotype, and no aberrations detected). This was rather a pilot study and based on these data we plan to further expand it in the next projects.

(2)   The authors state that the MALAT1 region is “not completely amplified in any patient…..so its expression function is questionable.” This is well shown in Fig 1.

By this sentence, we meant that since MALAT1 was partially amplified in the adenoma tissue, it does not necessarily mean that the transcribed MALAT1 lncRNA performs its function correctly. For example, lncRNA MALAT1 shares with miRNA the same recognition elements (MREs) to interact with them. If these MREs were not transcribed, lncRNA MALAT1 function would be defective independently of its gene amplification.

(3)   It is unclear as to whether patients were screened for MMR gene mutations and accessed for hypermutator phenotype.  

We thank the Reviewer for this interesting point. Adenomas are, after colonoscopy removal, sent for a histological examination to exclude a presence of malignancy. If assessed as benign, no additional screening is done including MMR genes. In our case, all adenomas were described by pathologists as a precancerous stage. We assume, that if there would be MMR gene mutations it would affect cells to such an extent that it would be reflected in the morphology that would be captured even by HE staining. However, from our previous work, only a few adenomas from the caecum and ascendant colon bear MMR mutations and MSI. The vast majority is MSI-L, e.g., from transversum down to the rectum (doi.org/10.1038/s41598-022-06498-9).

(4)   It would be useful to include a measure of polyp size.

We thank the Reviewer for this comment. We have included these data in Table 4 to complete information about adenoma samples.

Further experiments could strengthen the MS

(1)   The question of MALAT1 function would be approached by qRT-PCR and/or in situ hybridisation to demonstrate MALAT1 expression is increased compared to normal colon mucosa.

We agree with the Reviewer’s opinion. Unfortunately, we do not have on our disposal RNA in these patients. Additional isolation is not possible because a whole piece of fresh frozen tissue samples was already used for DNA isolation, since the aCGH method requires minimal input of 500ng of good quality DNA. For the same reason, we cannot perform FISH. Besides, the fresh frozen biopsies of adenomas consist of a tiny amount of material.

(2)   DNA sequencing would demonstrate MALAT1 amplification within the five group 1 polyps.

Indeed, DNA sequencing would provide the following supportive evidence of MALAT1 amplification. However, we are limited by biological material and sufficient high-quality DNA is required for next-generation sequencing. Nevertheless, we will of course include MALAT1 into the DNA panel for future planned research on new colorectal adenomas.